# Quantifying interactions in the active encounter complex of frustrated Lewis pairs

Alastair T. Littlewood [1], Tao Liu [2], Laura E. English [1], Linjiang Chen [1], Timothy A. Barendt [1] ✉ & Andrew R. Jupp [1] ✉

Sustainable catalysts based on main-group elements, such as frustrated Lewis pairs (FLPs), have emerged as alternatives to precious metal systems. The initial reaction of the Lewis acid, Lewis base and small molecule (*e.g.* $H_2$) is formally termolecular, but the reaction is rationalised by the pre-association of the acid and base in an encounter complex. Here we show that the charge-transfer band between $P(mes)_3$ and $B(C_6F_5)_3$ can be analysed by supramolecular UV-vis spectroscopic techniques to provide the key thermodynamic parameter, the association constant ($K_a$), for the active encounter complex, *i.e.* the pre-associated complex that is specifically in the correct orientation for small-molecule activation. We also demonstrate that a higher concentration of active encounter complex in solution leads to a faster activation of hydrogen. This method enables researchers to directly probe the complex that underpins FLP small-molecule activation and subsequent catalysis, and will aid the design of more active sustainable catalysts.

There is a huge drive to develop new sustainable chemical reactions fit for the 21st century. Frustrated Lewis pairs (FLPs) have emerged as a versatile class of main-group catalysts for a wide range of reactions and applications[1–6]. The original definition of FLPs described them as systems that comprise bulky Lewis acids and bases that are precluded from forming a Lewis adduct, though there are several examples of systems where there is a significant dynamic interaction between the donor and acceptor orbitals of the Lewis base and acid, respectively[7–10]. The latent reactivity of these unquenched acidic and basic sites can be exploited for the cooperative activation of small molecules, including the heterolytic cleavage of dihydrogen, $H_2$. The resulting proton and hydride can subsequently be delivered to a wide range of unsaturated organic substrates, such as alkenes, imines, and ketones, to promote the catalytic reduction of these functional groups without the need for precious metals (Fig. 1)[11–13]. The scope of FLP chemistry continues to grow, with applications in C–H activation[14], asymmetric catalysis[15,16], heterogeneous catalysis[17–19], and polymer synthesis[20,21].

The first step in the catalytic cycle of FLP hydrogenation is the splitting of $H_2$ by the FLP (Fig. 1). For archetypal intermolecular FLPs, this step involves the apparent simultaneous collision of three distinct molecules: the Lewis acid, the Lewis base, and $H_2$. The rationale to

explain this termolecular reactivity is the pre-association of two of the components. For $H_2$ activation by a phosphine and a borane, it has been shown that the Lewis acid and base form a weakly-bound species called the "encounter complex", which features a reactive pocket into which a molecule of $H_2$ can diffuse[22]. The encounter complex was first proposed by Pápai and co-workers in a computational study, where they identified a weakly associated $[P(^tBu)_3]\cdots[B(C_6F_5)_3]$ adduct as a minimum on the potential energy surface[23]. They showed that the encounter complex is not held together by a classical P→B dative bond, but instead by a large number of individually weak C–H···F non-covalent interactions. These C–H···F interactions for a range of FLP systems were corroborated by non-covalent interaction (NCI) analysis[24,25]. The favourable stabilisation energy in FLP systems is significant (-10–15 kcal mol⁻¹), and the inclusion of implicit solvent corrections only slightly reduces this stabilisation[26]. However, the enthalpic stabilisation is opposed by the entropic cost of adduct formation, and this is consistent with difficulties in observing the encounter complex experimentally[22,27].

Compelling evidence for its formation in solution was provided by ¹⁹F,¹H HOESY (Heteronuclear Overhauser Enhancement Spectroscopy) experiments performed on concentrated (220–230 mM) samples of

¹School of Chemistry, University of Birmingham, Edgbaston, Birmingham B15 2TT, UK. ²Department of Chemistry, University of Liverpool, Liverpool L69 7ZD, UK. ✉e-mail: t.a.barendt@bham.ac.uk; a.jupp@bham.ac.uk

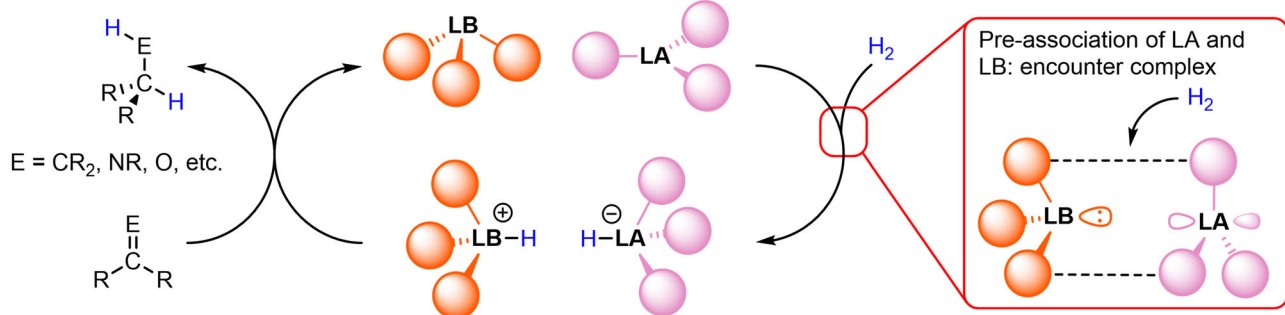

**Fig. 1 | General catalytic cycle for FLP hydrogenation catalysis.** This cycle features a generic three-coordinate Lewis base (LB) and Lewis acid (LA), and highlights the crucial role of the encounter complex stabilised by dispersion interactions.

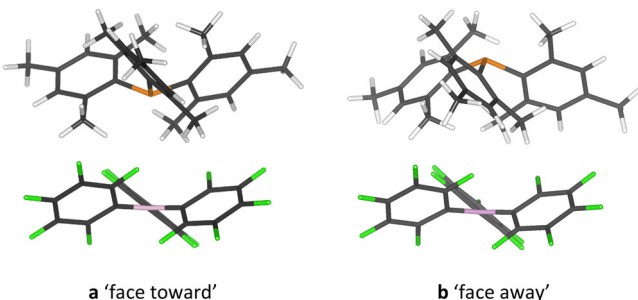

**Fig. 2 | Optimised structures of the two extreme forms of the encounter complex comprising P(mes)₃ and B(C₆F₅)₃. a** The "active encounter complex" with the phosphine lone pair pointing directly towards the borane p orbital. **b** The non-reactive encounter complex with the phosphine lone pair pointing away from the borane p orbital. Calculations were carried out at the B97D3(BJ)/Def2-SVP level of theory.

P($^{t}$Bu)$_3$/B(C$_6$F$_5$)$_3$ or P(mes)$_3$/B(C$_6$F$_5$)$_3$ (mes = mesityl, 2,4,6-trimethyl-phenyl) in toluene or benzene, where cross-peaks corresponding to H···F interactions could be observed[28]. Interestingly, the data showed that there are interactions between all fluorine environments on the borane and all proton environments on the phosphine, indicating that the two components are randomly oriented within the encounter complex in solution. The lack of preference for a particular configuration was supported by further high-level computations, which showed that the two "extreme" configurations shown in Fig. 2 are very similar in energy (<1 kcal mol$^{-1}$ difference) for P(mes)$_3$/B(C$_6$F$_5$)$_3$[29]. There is currently no method that can probe the "active encounter complex" (Fig. 2a) in solution, i.e. where the Lewis acid and base components are oriented in the correct manner for small-molecule activation and subsequent catalysis.

Supramolecular chemistry has developed the tools with which to study interactions between molecules, an important aspect being to quantify the binding strength of a non-covalent complex by measuring its association constant, $K_a$. Knowledge of $K_a$ informs the design of supramolecular complexes with tuneable affinities, which has proven critical for applications across chemical sensing[30], sequestration[31], and catalysis[32]. The only previous attempt to quantify the $K_a$ of an FLP encounter complex experimentally was that of P(mes)$_3$/B(C$_6$F$_5$)$_3$ in deuterated benzene; the method consisted of using diffusion ordered $^1$H and $^{19}$F NMR spectroscopy (DOSY) to determine the hydrodynamic radii for a small series of P(mes)$_3$:B(C$_6$F$_5$)$_3$ ratios[28]. These hydrodynamic radii were compared to the free species (P(mes)$_3$ and B(C$_6$F$_5$)$_3$) to predict a mole fraction of the encounter complex and ultimately an estimation of $K_a = 0.5 \pm 0.2$ M$^{-1}$. Alongside a number of assumptions to approximate the hydrodynamic radii, this approach required an adapted NMR single-point method to estimate $K_a$, which is

considered to be less accurate than fitting a supramolecular titration curve through non-linear regression[33]. A recent article determined the association constants of dispersion-stabilised Lewis pairs comprising phosphines and boranes, but these adducts featured P−B dative bonds and are thus fundamentally different to the "frustrated" system described here[34]. Furthermore, an elegant study using microwave dielectric spectroscopy to assess the non-covalent interaction of acids and bases in solution has recently been published, though it is not possible to determine association constants from these data[35].

We quantify FLP association using a UV-vis spectroscopic titration, a methodology that enables us to determine $K_a$ by accurate non-linear curve fitting[36], the gold standard in supramolecular chemistry[33]. This strategy provides an experimental probe to quantify the association constant of an FLP "active encounter complex" in solution, an outcome that is supported by an extensive computational study into the relative orientations of P(mes)$_3$ and B(C$_6$F$_5$)$_3$ in the encounter complex. We have used these results to show that a higher concentration of the active encounter complex in solution leads to faster small-molecule activation. The fundamental understanding of the encounter complex we uncover here will facilitate the design of more active FLPs.

## Results

P(mes)$_3$ and B(C$_6$F$_5$)$_3$ are both colourless when independently dissolved in toluene, but the combination of the two gives rise to a magenta colour (see Fig. 3 and S1, S2). This fact has been documented since the earliest report of this system being used as an FLP[37], but it was only recently determined that this colour arises from the formation of a charge-transfer complex between the Lewis acid and base[38,39], and not from the build-up of radicals[40]. This was proved by measuring the EPR spectrum of a freshly prepared sample of the magenta solution of P(mes)$_3$ and B(C$_6$F$_5$)$_3$ in toluene and showing there were no resonances for the radicals, but after irradiation of the sample at 534 nm inside the spectrometer large signals for the frustrated radical pair could be seen[39]. The charge-transfer band with $\lambda_{max} = 534$ nm enabled us to directly measure the association constant ($K_a$) of the Lewis acid and base of an FLP in a toluene solution (Fig. 3a). The P(mes)$_3$/B(C$_6$F$_5$)$_3$ complex is air- and moisture-sensitive; the pale magenta colour in solution begins disappearing immediately upon exposure to standard atmospheric conditions (Fig. S35). This necessitated that all samples were prepared and analysed after equilibration in an N$_2$-filled glovebox.

To determine $K_a$[36], 13 discrete solutions of P(mes)$_3$/B(C$_6$F$_5$)$_3$ were prepared under inert conditions with a constant concentration of the Lewis acid (5 mM), and increasing concentrations of Lewis base (up to 300 mM, i.e. 60 equivalents). As expected, increasing the Lewis base:acid ratio led to an increase in intensity of the magenta colour, which is clearly visible to the naked eye (Fig. 3b). The intensity of the charge-transfer absorption band at $\lambda_{max} = 534$ nm was measured by

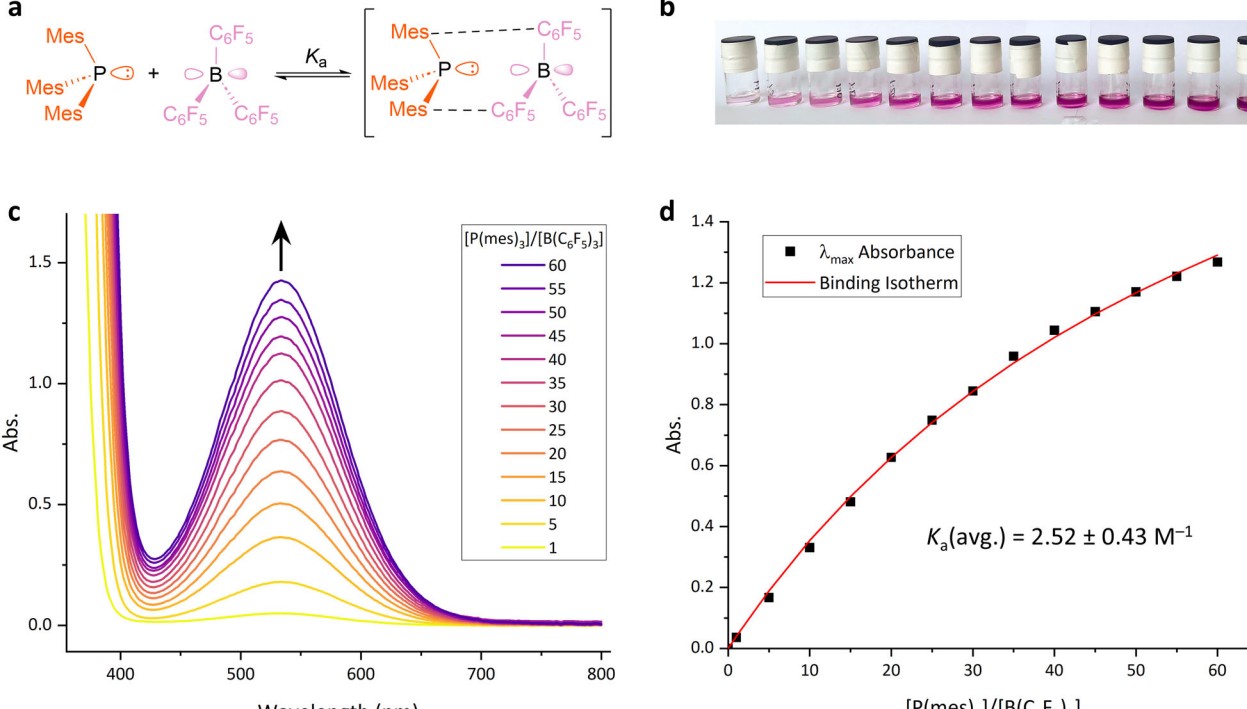

**Fig. 3 | Analysis of the charge-transfer band for the P(mes)$_3$/B(C$_6$F$_5$)$_3$ system.**
**a** Complexation for which $K_a$ is being determined. **b** Solutions used in titration experiments; increasing magenta colour with increasing ratios of phosphine:borane. **c** Increasing intensity of charge-transfer band with increasing Lewis base concentration. **d** One of three plots of $\lambda_{max}$ absorbance as a function of phosphine:borane ratio, and the binding isotherm from the fit of these data, and the average $K_a$ obtained from the three experiments.

UV-vis spectroscopy (Fig. 3c), and showed the characteristic increase before beginning to level off at higher concentrations of phosphine (Fig. 3d). The $K_a$ value was calculated by non-linear fitting of the resulting titration curve to a 1:1 stoichiometric binding model[33,41]. Titrations were carried out in triplicate (with separate samples prepared each time), to give an average $K_a$ of 2.52 M$^{-1}$ with a relative standard deviation of 0.43 M$^{-1}$ for the FLP P(mes)$_3$/B(C$_6$F$_5$)$_3$. This average value corresponds to $\Delta G = -0.55$ kcal mol$^{-1}$ for the association of the acid and base in toluene at 301 K; this low value is consistent with previous computational reports that the encounter complex will only be present in low concentration in solution[24,26,29], although it does highlight that association is a very slightly exergonic process.

There are many possible orientations of the associated acid and base within the encounter complex, and previous experimental efforts to characterise the encounter complex have only been able to show that the two components are associated, but without any differentiation between the relative orientations[22]. Many of these orientations could generate a charge-transfer band and thus enable single-electron transfer (SET) due to the overlap of various donor and acceptor orbitals on the phosphine and borane, respectively, such as π–π* transitions due to π-stacking interactions of the aryl rings on the phosphine and borane.

We hypothesised that our UV-vis spectroscopic probe is sensitive only to the active encounter complex, i.e. the orientation that is correctly set up for small-molecule activation. Our hypothesis was based on the fact that the HOMO of the P(mes)$_3$/B(C$_6$F$_5$)$_3$ combination is predominantly the lone pair on P, and the LUMO is the formally vacant p orbital on B[39]. Therefore, alignment of these frontier orbitals would presumably permit the lowest energy SET process, which would correlate to the charge-transfer absorption band monitored in the supramolecular titration (Fig. 3c). We, therefore, sought to probe this chemical space in a systematic and rigorous manner to determine the source of our diagnostic charge-transfer band.

We explored a wide range of possible orientations for the acid and base within the encounter complex, beyond the two extremes in Fig. 2. The input coordinates for the P(mes)$_3$/B(C$_6$F$_5$)$_3$ encounter complexes were generated in two ways (grid search and scanning method), and 1644 different permutations were used to explore the chemical space as comprehensively as possible (Fig. 4a); full details can be found in the SI. All 1644 input orientations were fully geometry-optimised, and their energies were calculated using the semi-empirical GFN2-xTB method (Fig. 4b)[42]. By applying a cut-off of 5 kcal mol$^{-1}$ above the global energy minimum on the xTB binding energy landscape (Fig. 4b), 810 P(mes)$_3$/B(C$_6$F$_5$)$_3$ binding configurations were selected and further geometry-optimised using density functional theory (DFT: B97D3(BJ)/Def2-SVP)[43–45], of which 774 converged successfully. The binding energies of the optimised 774 configurations were determined by single-point energy calculations at the ωB97XD/Def2-TZVP level of theory[45,46], and this binding energy landscape is shown in Fig. 4c. These data show that there are many energetically accessible orientations, consistent with the aforementioned NMR spectroscopy experiments by Rocchigiani et al. [28]. Reassuringly, the binding energies of the two extreme orientations (Fig. 2) are very similar (−10.285 and −10.287 kcal mol$^{-1}$), in good agreement with the previous calculations performed by Grimme and co-workers[29].

The lowest energy transitions within these different optimised orientations were calculated using time-dependent DFT (TD-DFT). The lowest energy orientation from each cluster was used to create a set of 72 representative configurations, and the vertical excitation for each configuration was simulated at the TD-M062X/Def2-SVP level of theory[45,47]. These data allowed us to compare the three separate criteria: binding energies, S$_1$ transition energies, and respective oscillator strengths (Fig. 4d).

The only combination of the three parameters that is consistent with the experimentally observed transition is the point depicted in Fig. 4e, which features the transition with the lowest S$_1$ transition

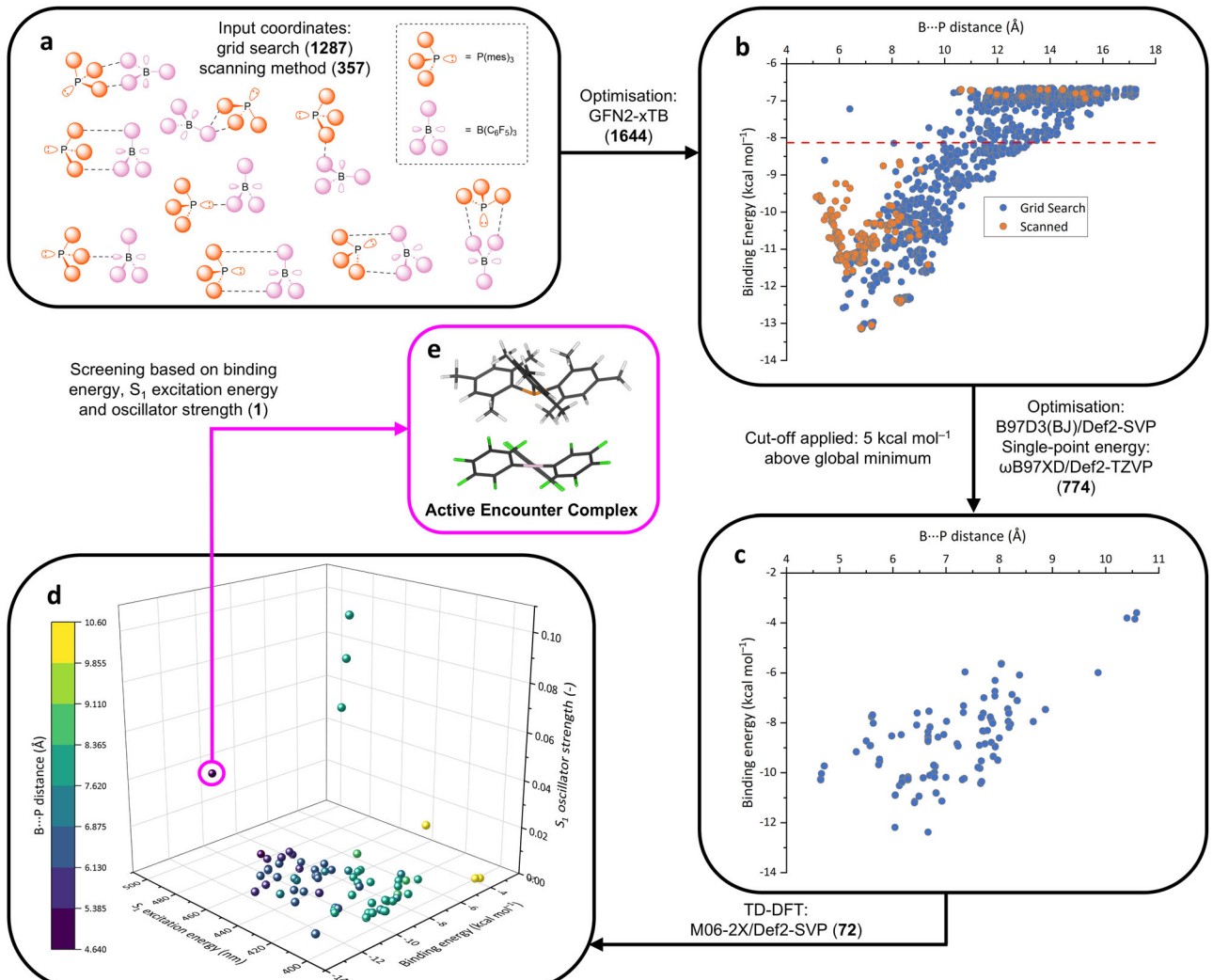

**Fig. 4 | Flowchart of computational study on the active encounter complex.**
The numbers in bold show how many configurations are being carried through the workflow at each stage. **a** Schematic representations of some of the 1644 input coordinates of the encounter complexes. **b** Graph showing binding energy as a function of P···B distance calculated using a semi-empirical method. The cut-off of 5 kcal mol$^{-1}$ above the global energy minimum is shown by a red dashed line, and all points above this line were discarded. **c** Graph showing binding energy as a function of P···B distance, calculated using DFT; the 774 data points have converged to 72 distinct clusters. **d** Three-dimensional plot comparing the binding energy with the S$_1$ excitation energies and oscillator strengths from TD-DFT. The range of P···B distances for each point is also indicated by the colour chart. **e** The structure of the only data point that has values from the plot in d that is consistent with the experimentally observed absorbance band; the active encounter complex.

energy (499 nm, in reasonable agreement with the experimental value of 534 nm), a relatively significant oscillator strength, and a binding energy close to the global minimum. This data point corresponds to the orientation where the phosphine lone pair is pointing directly at the formally vacant p orbital on the borane, i.e. the orientation depicted in Fig. 2a. The majority of the calculated transitions (57 of the 72) have negligible oscillator strengths (<0.005) and thus these orientations are highly unlikely to contribute to the experimental absorbance band. The three transitions with the highest oscillator strengths (labelled α, β and γ in Figs. S48, S49) all arise from similar orientations that feature π-stacking interactions between one mesityl ring on the phosphine and one C$_6$F$_5$ ring on the borane, but these three orientations are relatively higher in energy (−8.02, −7.87 and −7.77 kcal mol$^{-1}$, respectively) than the other orientations and are thus also unlikely to have any appreciable contribution to the absorbance band (where the binding energy of the 'face towards' complex is −10.29 kcal mol$^{-1}$).

To probe entropic contributions, a cut-off at −10 kcal mol$^{-1}$ in Fig. 4c was applied, and 13 P(mes)$_3$/B(C$_6$F$_5$)$_3$ binding conformations

were therefore selected and further investigated using the M06/6-311 G(2df,p) level of theory with the D3 version of Grimme's dispersion correction[48]. As shown in Fig. S46, the entropic effect decreases the stability of the associated phosphine/borane by -20 kcal mol$^{-1}$ (ranging from −18 to −23 kcal mol$^{-1}$), resulting in ΔG values ranging from −3 to 2.5 kcal mol$^{-1}$ (at 301 K). Six structures exhibit ΔG < 0, with the 'face toward' configuration (Fig. 2a) having the lowest ΔG, though it is only 1.8 kcal mol$^{-1}$ more stable than the other five configurations. This suggests that the 'face toward' configuration co-exists with other configurations in solution, with no single dominant orientation, although it is significant that the active encounter complex in the P(mes)$_3$/B(C$_6$F$_5$)$_3$ FLP is the most stable orientation. This result reveals that our UV-vis spectroscopic titration method enables us to quantify the active encounter complex in solution, experimentally providing the key thermodynamic parameter of the weakly associated species in solution that underpins FLP small-molecule activation and subsequent catalysis.

Knowledge of the $K_a$ of the active encounter complex enables accurate mole fractions to be determined; for example, using the

**a**

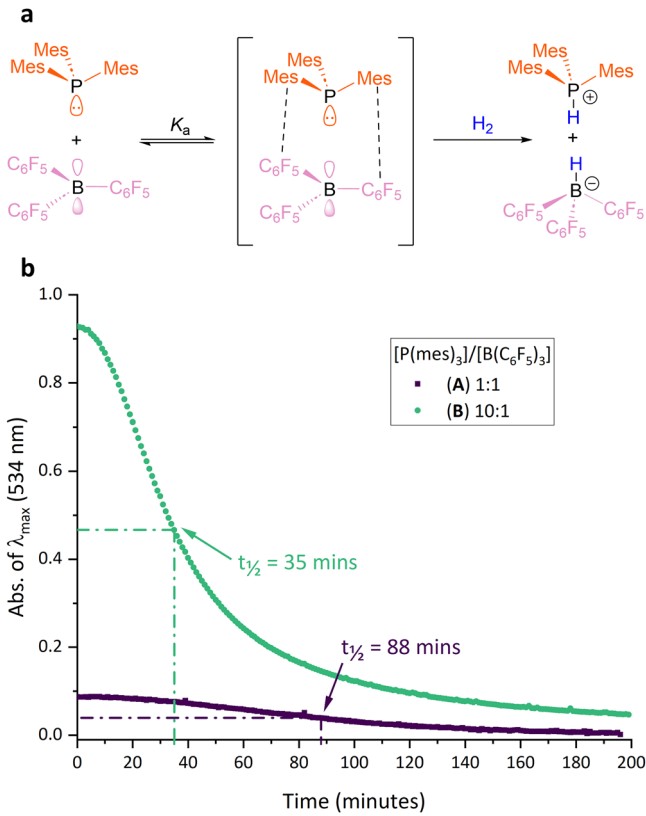

**b**

Fig. 5 | Rate of hydrogen uptake observed through the quenching of the
P(mes)$_3$/B(C$_6$F$_5$)$_3$ charge-transfer band. a H$_2$ activation by FLP being studied.
b Change in λ$_{max}$ (534 nm) absorbance of 1:1 (purple squares – reaction **A**) and 10:1
(green circles – reaction **B**) ratios of P(mes)$_3$/B(C$_6$F$_5$)$_3$ under a flow of 5% H$_2$ and 95%
N$_2$ as a function of time.

average $K_a$ of 2.52 M$^{-1}$, a 1:1 mixture of P(mes)$_3$/B(C$_6$F$_5$)$_3$ at 5 mM con-
centration in toluene (which matches the concentration of our UV-vis
titrations) would result in 1.2% of the components being in the active
encounter complex at any moment. This low percentage could par-
tially explain why FLP catalysts have relatively low activity compared to
transition metal systems, as in this case 98.8% of the catalyst is not in its
active form at any one time. But by increasing the relative ratio of
P(mes)$_3$ to B(C$_6$F$_5$)$_3$ to 10:1, 11.1% of the limiting borane would be in the
active encounter complex in solution. It is therefore possible to sig-
nificantly increase the amount of active catalyst by simply increasing
the amount of only one component (the Lewis base in this case), which
also has economic advantages as the Lewis acid B(C$_6$F$_5$)$_3$ is typically the
more expensive component within the FLP.

We sought to experimentally verify this prediction by studying the
rate of hydrogen activation by the FLP as a function of encounter
complex concentration (Fig. 5a). Two reactions were set up with
P(mes)$_3$/B(C$_6$F$_5$)$_3$ ratios of 1:1 (**A**) and 10:1 (**B**), respectively, in toluene
with the B(C$_6$F$_5$)$_3$ at 5 mM concentration, and a constant flow of H$_2$ was
passed over the stirring reactions. As the FLP reacted with hydrogen to
form the colourless salt [HP(mes)$_3$][HB(C$_6$F$_5$)$_3$], the magenta colour
from the P(mes)$_3$/B(C$_6$F$_5$)$_3$ encounter complex became less intense. To
make the kinetics of the reactions easier to follow, 5% H$_2$ in 95% N$_2$
carrier gas was used to slow down the rate of hydrogen activation. The
intensity of the absorption band at 534 nm was monitored over time
for the two reactions. The results in Fig. 5b clearly show that the loss of
intensity is much faster for **B** than **A**, i.e. when there is a higher initial
concentration of active encounter complex. In **A**, the reaction takes
88 min for the absorption band to be at half its starting intensity,
whereas this only takes 35 min for **B**; note this includes an induction
period for the hydrogen to diffuse into the solution. Subsequent $^{31}$P,

$^{19}$F, and $^{11}$B NMR analysis on samples from **A** and **B** was undertaken to
corroborate the expected hydrogen-activated products.

To quantify the extent of hydrogen activation by the FLP, reac-
tions **A** and **B** were repeated and then stopped after 60 min by evac-
uating the reaction flask to prevent further reactivity. After removal of
the toluene, the reaction samples were dissolved in CDCl$_3$ and ana-
lysed by quantitative $^{31}$P NMR spectroscopic experiments (see Figs.
S31, S33), which revealed that 25% of the P(mes)$_3$/B(C$_6$F$_5$)$_3$ had been
converted to [HP(mes)$_3$][HB(C$_6$F$_5$)$_3$] in **A** after an hour, whereas the
value was 41% for **B** (relative to the limiting reagent B(C$_6$F$_5$)$_3$). These
experiments clearly show that a significant enhancement in reaction
rates can be obtained by increasing the concentration of active
encounter complex in solution. The resulting data underpin previous
hypotheses for the increased rate of reaction of FLPs where there is an
excess of Lewis base relative to Lewis acid due to increased con-
centration of the encounter complex[49,50]. These findings also corro-
borate previous studies on the auto-induced catalytic hydrogenation
of imines and imidoyl chlorides that show an increase in the rate of
conversion as more of the Lewis basic amine is produced[51–53], although
using our methodology we are able to focus solely on the fundamental
step of hydrogen activation.

This method for determining the $K_a$ for FLP systems is predicated
on the presence of an appropriate charge-transfer band, so we assessed
how these charge-transfer bands varied with the extent of methylation
on the triarylphosphines in combination with B(C$_6$F$_5$)$_3$ in toluene. It is
known that triphenylphosphine (PPh$_3$) creates a Lewis adduct with
B(C$_6$F$_5$)$_3$, which precludes the formation of a charge-transfer band[54,55].
Tris(*ortho*-tolyl)phosphine (P(*o*-tol)$_3$; **i** in Fig. 6a) affords a faint yellow
colour on mixing with B(C$_6$F$_5$)$_3$, although the charge-transfer band
partially overlaps with the absorption of B(C$_6$F$_5$)$_3$, hampering further
analysis of this band. In contrast, there were clear charge-transfer
bands for combinations of B(C$_6$F$_5$)$_3$ with tris(2,6-dimethylphenyl)
phosphine (P(xyl)$_3$; **ii**, λ$_{max}$ = 451 nm), tris(2,3,5,6-tetramethylphenyl)
phosphine (P(dur)$_3$, **iv**, λ$_{max}$ = 496 nm), and tris(pentamethylphenyl)
phosphine (P(C$_6$Me$_5$)$_3$, **v**, λ$_{max}$ = 500 nm). Promisingly, these charge-
transfer bands also show a clear increase in intensity as a function of
increasing phosphine:borane ratio (Fig. S4); this change is exemplified
by the P(dur)$_3$/B(C$_6$F$_5$)$_3$ system shown in Fig. 6b. However, it is note-
worthy that the charge-transfer bands for these four additional phos-
phines with B(C$_6$F$_5$)$_3$ are all significantly weaker than the analogous
band for P(mes)$_3$/B(C$_6$F$_5$)$_3$ (**iii**); all the spectra in Fig. 6a were carried out
at a 10:1 ratio of phosphine:borane, and the largest charge-transfer
band is clearly seen for P(mes)$_3$/B(C$_6$F$_5$)$_3$ despite this sample being run
at a lower concentration. The higher concentrations for the systems
involving **i**, **ii**, **iv** and **v** noted in Fig. 6a were, therefore, necessary to
unambiguously observe and analyse each charge-transfer band (see
Fig. S4). Unfortunately, this means that, using our current experimental
set-up, we were unable to accurately fit the titration data and calculate
$K_a$ values for these systems because the higher concentration of
phosphine required to observe the charge-transfer band with our
current spectrometer meant we reached the solubility limit of phos-
phine before we could collect enough titration data points at higher
stoichiometric ratios for accurate analyses. This study does provide
evidence that charge-transfer bands in FLP systems may be more
common than previously expected, and higher concentrations of
phosphine:borane stoichiometric ratios are required to observe and
study them. The physical limitation on the solubility of the phosphine
could be overcome in future by using a more sensitive UV-vis spec-
trometer or a cuvette with a longer path length, and we are actively
exploring both these possibilities.

## Discussion

We have developed a new methodology based on UV-vis spectroscopy
to directly probe the encounter complex in frustrated Lewis pair
chemistry, using the prototypical P(mes)$_3$/B(C$_6$F$_5$)$_3$ combination as an

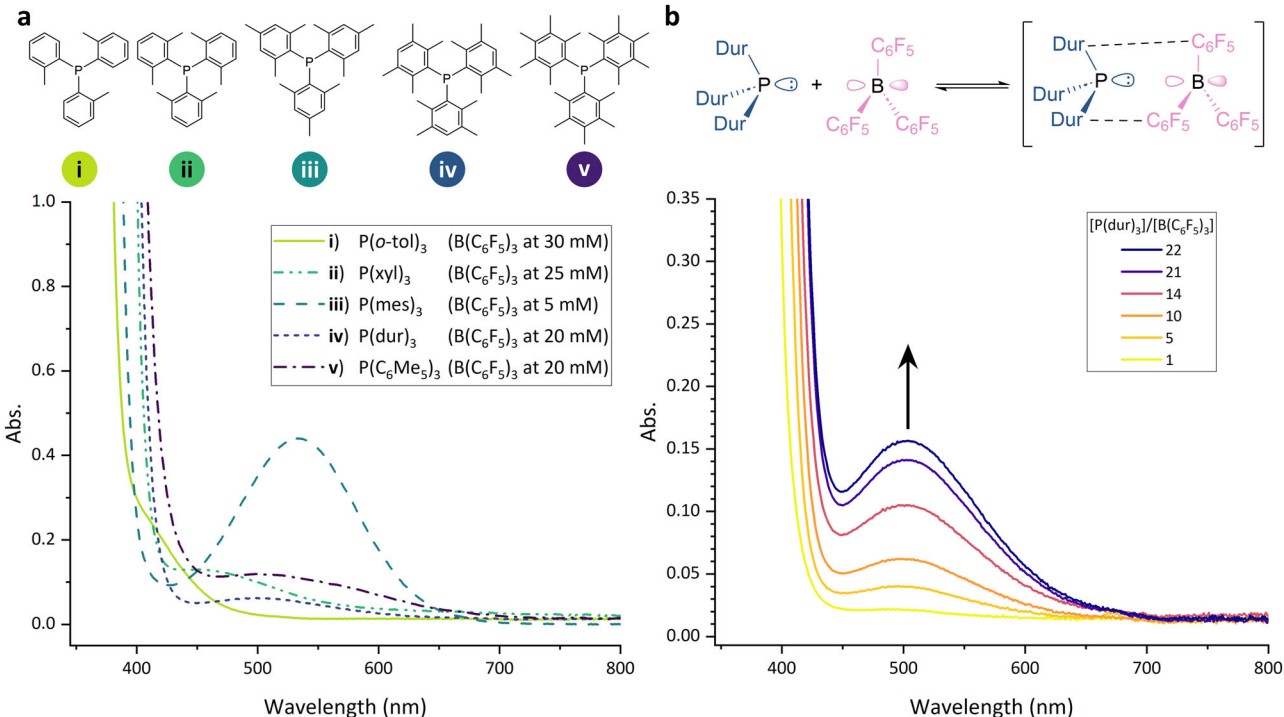

**Fig. 6 | Additional charge-transfer bands observed with differing triarylphosphines. a** Increasing methylation of the basic triarylphosphines and their associated CT bands when mixed with $B(C_6F_5)_3$ in a 10:1 (P:B) ratio, **i:** $P(o\text{-tol})_3$, **ii:** $P(xyl)_3$, **iii:** $P(mes)_3$, **iv:** $P(dur)_3$, **v** $P(C_6Me_5)_3$. **b** The encounter complex formed between $P(dur)_3$ and $B(C_6F_5)_3$ with the increase in $[P(dur)_3]$ leading to an increase in absorbance.

exemplar. We have employed best-practice techniques from supramolecular chemistry to determine the $K_a$ of $P(mes)_3/B(C_6F_5)_3$ in toluene of $2.52\,M^{-1}$, which shows that the association of the Lewis acid and base is slightly favourable under these conditions. The in-depth computational study has thoroughly explored the chemical space of this encounter complex; starting with 1644 different input orientations, and using increasingly high-level computational methods we were able to show that the only configuration that is consistent with the charge-transfer absorbance band used in our spectroscopic titration is the orientation where the phosphorus lone pair is pointing at the formally vacant p orbital on boron. As this orientation is the same as that required for small-molecule activation, our methodology enables an assessment of the key thermodynamic parameter of the active encounter complex that underpins FLP small-molecule activation and catalysis. We used the knowledge of the association constant and, therefore, the mole fraction of active encounter complex in solution to show that a higher concentration of active encounter complex leads to a faster rate of hydrogen activation. In addition, we have identified a further four triarylphosphines that, upon mixing in solution with $B(C_6F_5)_3$, form a charge-transfer band visible in the UV-vis spectrum that increases in intensity as a function of the phosphine:borane ratio, which opens the door to the quantitative investigation of their active encounter complexes by UV-vis spectroscopy. Research is currently on-going to further develop this methodology to assess the effects of experimental conditions on the formation of the active encounter complex, including different solvents, temperatures, and different FLP combinations and ratios. We anticipate this will enable the community to design more active main-group catalysts.

## Methods

### Association constant determination of $P(mes)_3/B(C_6F_5)_3$ pair

Stock solutions of 380 mM $P(mes)_3$ and 100 mM $B(C_6F_5)_3$ were made respectively using dried and degassed toluene and stored in air-tight ampoules. The desired ratios from 1:1 to 60:1 ($[P(mes)_3]:[B(C_6F_5)_3]$) were made up in separate vials (maintained at 5 mM $B(C_6F_5)_3$), with a total volume of 1 mL per sample. For the data collection, each sample was analysed within the glovebox on a Biochrom UV-vis spectrometer. The association constant ($K_a$) was determined by multiwavelength, non-linear curve fitting using a 1:1 model in BindFit[33]. The spectral region $\lambda = 524–544\,nm$ was used because this is where the largest changes in the charge transfer absorption band occurs and this region has no other conflicting absorbances. This procedure was carried out in triplicate.

### Titration of $P(dur)_3/B(C_6F_5)_3$ pair

Stock solutions of 250 mM $P(dur)_3$ and 500 mM $B(C_6F_5)_3$ were made respectively using the same source of dried and degassed toluene. The desired ratios from 1:1 to 22:1 ($[P(dur)_3]:[B(C_6F_5)_3]$) were made up in separate vials (maintained at 20 mM $B(C_6F_5)_3$), with a total volume of 200 µL per sample. For the data collection, each sample was sealed in a 100 µL cuvette with a Suba-Seal® septa and electrical tape to permit analysis on Cary-60 spectrometer.

### Hydrogen activation general procedure

Two FLP ratios, 1:1 and 10:1 ($P(mes)_3/B(C_6F_5)_3$), were formed in 5 mL of toluene ($B(C_6F_5)_3$ maintained at 5 mM) within a 15 mL ampoule. The solutions were stirred at 300 rpm, with a flow of 5% $H_2$ in $N_2$ gas (-0.1 bar on the regulator) and a relief needle attached to a bubbler to ensure gas flow was maintained throughout. A UV-vis absorbance measurement was taken every minute until the $\lambda_{max}$ absorbance was quenched to a point where the absorbance reading was no longer changing, ~200 min. The experiments were repeated for exactly 60 min before the ampoule was switched to vacuum to evacuate any residual $H_2$. The toluene was removed in vacuo, and both samples were redissolved in 0.7 mL $CDCl_3$ for NMR spectroscopic analysis.

## Synthesis of tris(2,6-dimethylphenyl)phosphine)

A freshly prepared solution of 2,6-dimethylphenylmagnesium bromide in THF (49 mL, 0.91 M, 44.5 mmol, 3.2 eq.) was added dropwise, over 1 h, to a solution of $PCl_3$ (1.2 mL, 1.98 g, 13.8 mmol, 1 eq.) in 40 mL THF at −78 °C. The resulting solution was then allowed to warm to room temperature and then stirred overnight. The solvent was removed in vacuo, and the resulting solid was extracted with 125 mL toluene to give a pale yellow solution. Removal of the solvent *in vacuo* produced an off-white solid which was recrystallised from hexane. Yield: 3.30 g, 69%. [1]H NMR (400 MHz, $d_8$-toluene) δ 6.99-6.95 (m, 3H, Ar-C*H*), 6.85-6.82 (m, 6H, Ar-C*H*), 2.15 (s, 18H, C*H*$_3$). [31]P{[1]H} NMR (162 MHz, $d_8$-toluene) δ − 34.6. NMR data are consistent with literature values[56].

## Synthesis of tris(2,3,5,6-tetramethylphenyl)phosphine)

A freshly prepared solution of 2,3,5,6-tetramethylphenyl magnesium bromide in THF (39 mL, 0.77 M, 30.0 mmol, 3.3 eq.) was added dropwise, over 1 h, to a solution of $PCl_3$ (1 mL, 1.25 g, 9.1 mmol, 1 eq.) in 10 mL THF at −78 °C. The resulting solution was then allowed to warm to room temperature and then stirred overnight. The solvent was removed in vacuo and the resulting solid was extracted with 100 mL toluene to give a pale yellow solution. Removal of the solvent *in vacuo* produced an off-white solid which was recrystallised from hexane. Yield: 2.00 g, 51%. [1]H NMR (400 MHz, $d_8$-toluene) δ 6.88 (s, 3H, Ar-C*H*), 2.23 (s, 18H, C*H*$_3$), 2.07 (s, 18H, C*H*$_3$). [31]P{[1]H} NMR (162 MHz, $d_8$-toluene) δ − 29.1. NMR data are consistent with literature values[57,58].

## Synthesis of tris(pentamethylphenyl)phosphine)

A freshly prepared solution of pentamethylphenylmagnesium bromide in THF (59 mL, 0.25 M, 15 mmol, 3.3 eq.) was added dropwise, over 1 h, to a solution of $PCl_3$ (0.4 mL, 0.62 g, 4.5 mmol, 1 eq.) in 20 mL THF at −78 °C. The resulting solution was then allowed to warm to room temperature and then stirred overnight. The solvent was removed *in vacuo* and the resulting solid extracted with 150 mL hexane to give a pale yellow solution. Removal of the solvent *in vacuo* produced an off-white solid which was recrystallised from hexane. Yield: 0.66 g, 23%. [1]H NMR (300 MHz, CDCl$_3$) δ 2.25 (s, 9H, C*H*$_3$), 2.16 (s, 18H, C*H*$_3$), 2.07 (s, 18H, C*H*$_3$). [31]P{[1]H} NMR (121 MHz, CDCl$_3$) δ − 25.1. NMR data are consistent with literature values[59].

## Synthesis of tris(pentafluorophenyl)borane (B(C$_6$F$_5$)$_3$)

A freshly prepared solution of pentafluorophenylmagnesium bromide in Et$_2$O (200 mL, 0.6 M, 120 mmol, 3 eq.) was added dropwise to a vigorously stirred solution of BF$_3$·Et$_2$O (5 mL, 5.75 g, 40 mmol, 1 eq.) in 80 mL toluene at 0 °C. The reaction was then allowed to warm to room temperature, and the Et$_2$O was removed in vacuo. The resulting toluene solution was then heated to 98 °C for 1 h using a water bath before cooling to room temperature and removing the remaining solvent in vacuo. The product was extracted three times with warm hexane and crystallised by cooling the hexane solutions to −30 °C. Yield: 15.302 g, 65%. The analytically pure material is obtained by two consecutive sublimations under a dynamic vacuum ($1 \times 10^{-2}$ mbar) at 90 °C. Average sublimation yield: 85%. [11]B{[1]H} NMR (128 MHz, CDCl$_3$) δ 57.8 (br, s). [19]F{[1]H} NMR (376 MHz, CDCl$_3$) δ − 128.0 (s, 6 F), −143.0 (s, 3 F), −159.9 (m, 6 F). NMR data are consistent with literature values[60].

## Computational general procedure

For detailed information on the computational analyses carried out, please refer to the computational details section of the Supplementary Information.

## Data availability

All data generated in this study are available in the open access UBIRA database https://doi.org/10.25500/edata.bham.00001256. All data are also available from the corresponding author upon request.

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

## Acknowledgements

A.T.L., L.C., T.A.B. and A.R.J. acknowledge the University of Birmingham for funding. A.R.J. would also like to thank the Royal Society (URF \R1\201636), A.R.J and L.E.E. gratefully acknowledge the EPSRC (EP/ W036908/1), and T.A.B. acknowledges the EPSRC (EP/W037661/1) for funding. Dr Cécile Le Duff is gratefully acknowledged for NMR spectroscopy discussions.

## Author contributions

A.R.J. and T.A.B. designed the experiments. A.T.L. conducted the experiments. L.E.E performed the phosphine syntheses. A.T.L., T.A.B. and A.R.J. analysed experimental data. T.L. and L.C. conducted the computations. T.L., L.C and A.R.J analysed the computational data. A.T.L. and A.R.J. wrote the manuscript with input from all authors.

## Competing interests

The authors declare no competing interests.
