## [Transparent Peer Review file · Nature Communications]

Quantifying Interactions in the Active Encounter Complex of Frustrated Lewis Pairs

Corresponding Author: Dr Andrew Jupp

Version 0:

Reviewer comments:

Reviewer #1

(Remarks to the Author)

The manuscript of Jupp, Barendt and coworkers presents the experimental determination of K_a for the benchmark frustrated Lewis pair $\text{PMe}_3 / \text{BCF}$. As such, the work is an essential contribution to the field of main group chemistry and catalysis. Yet, as the novel findings concerns merely the establishment of K_a , the manuscript is imho not suited for the Nature portfolio.

My comments are:

-line 9: the authors claim precious metal systems are environmentally unfriendly. Can the authors support this claim with scientific findings or rephrase this sentence?

-I12: "acid, base" I would recommend using the Lewis acid, Lewis base nomenclature to be more precise.

-I14: "However, there is no experimental methodology to study the active encounter complex". The authors highlight literature examples that showcase NMR spectroscopy (I58, I79) and UV-vis spectroscopy (I89) has been used as an experimental tool to characterise and study the encounter complex in FLP chemistry. I suggest the authors rephrase this claim.

-I91: "This strategy provides the first experimental probe for the "active encounter complex" of an FLP in solution". I am not in favour of this claim given the cited literature examples on NMR and UV-vis spectroscopy. Can the authors rephrase this? The authors are indeed the first to quantify K_a , yet qualitative evidence for the encounter complex has been provided before.

-I130: "very slightly exothermic process" I suggest to replace exothermic with exergonic, which is more appropriate term for a change in Gibbs free energy.

-I201: "borane, but these three orientations are relatively higher in energy than the other orientations". Can the authors add the relative energies in the main text to provide the reader directly insight (e.g. Boltzmann) if the higher energy states are populated or not?

-general question: is the conformation associated with the CT band the only "active encounter complex in solution" or can other conformations (many are calculated) also directly engage in a productive reaction with substrates like dihydrogen? if yes, then the quantification of K_a is directly coupled to the concentration of the CT structure, yet a higher concentration of "active encounter complex in solution" is present.

-for hydrogen activation experiments "active encounter complex" concentrations are used of 1.2% (1:1; A) and 11.1% (10:1; B), which is a ~10 fold increase. The authors use two approaches to quantify the reaction progress. 1. The UV-vis determination of the unreacted (CT) encounter complex concentration, 2. The determination of the H_2 -activated product by ^{31}P , ^{19}F , and ^{11}B NMR analysis. After 60 min, quantitative ^{31}P NMR indicated 23% (A) vs 41% (B) was formed. So my question is: given the tenfold increase in active FLP concentration and, likely, second order kinetics ([active encounter complex][H_2]), can the difference in product yield be understood? Intuitively, I would expect a bigger difference in yield.

Reviewer #2

(Remarks to the Author)

Littlewood and co-authors have developed an experimental procedure that allows to determine the association constant for the active encounter complex of so-called frustrated Lewis pairs, with relatively high accuracy. They validated their results by a computational screening of all possible FLP complex configurations and funneling down to the only plausible candidate based on spectroscopic data, corresponding to the active complex.

This work is of high relevance to the field, since it gives experimental evidence of the existence of such pre-association of acid and base, as was already predicted based on computational grounds before. The authors also confirmed that the presence of these complexes is directly related to H₂ activation rates, thus indicating that tuning of the strength of association will impact the catalytic behavior.

The proposed methodology looks very sound, both experimentally and computationally. As it was shown before in literature that different orientations of acid and base can co-exist, the authors followed the right track by using grid search and scanning methods to identify all plausible configurations, first by semi-empirical method and later fine-tuned by DFT. However, what I was missing was the highly relevant entropic effect. When accounting for the entropic contribution, does the computational binding free energy still agree with the experimentally determined exergonic association process? In literature, for this particular FLP, not all studies agreed on its favorability. Could the authors give an estimation of the Gibbs free binding energy, or at least comment on it? Somewhat related to that, can the authors clearly mention in the main manuscript at what temperature the experiments were conducted?

Finally, it is a pity that only a single FLP was considered for the study. If not in this work, it would be nice to see whether trends in computationally calculated strengths are conserved with this experimental approach.

After addressing the few remarks raised above, I recommend publication of this manuscript in Nature Communications.

Reviewer #3

(Remarks to the Author)

Review of the manuscript "The Active Encounter Complex in Frustrated Lewis Pair Chemistry" by Alastair T. Littlewood, Tao Liu, Linjiang Chen, Timothy A. Barendt, Andrew R. Jupp, submitted to Nature Communications

The authors present a very accurate determination of the equilibrium constant K_a using sophisticated, state-of-the-art UV/VIS titration techniques including a triple determination for the association of probably the most studied FLP system in the literature, B(C₆F₅)₃ and PMes₃. Quantum chemical calculations were used to complement the absorption measurements. The employed methodology is sound even if the computational protocol could still be improved (but this would most likely not change the qualitative conclusions). The work can be fully reproduced by the provided details and data in the manuscript/supporting information and the data interpretation is precise.

Detailed comments:

Especially the UV/VIS part of the work is excellent, but the applicability of this approach is somewhat limited to those FLPs that also show UV/VIS absorption. This raises the question of how many of the (active) FLPs actually do so. The authors should add an analysis in order to be able to evaluate what impact the presented approach could actually have in practice in the search for better FLPs etc. (if necessary, the statement would then have to be weakened somewhat). Nevertheless, this work is very valuable, as there are practically no reliable methods to characterize these flexible systems in solution (even if the sentence in the abstract "However, there is no experimental methodology to study the active encounter complex, ..." should be somewhat weakened (the authors mention also other experiments in the paper, e.g. using DOSY).

Likewise here: Page 3, lines 91-92 "This strategy provides the first experimental probe for the "active encounter complex" of an FLP in solution,..."

When I do a quick literature search, I come across other experimental approaches to characterize the active encounter complex, see e.g. <https://doi.org/10.1039/D2RE00123C>. Here the authors should use weaker formulations.

The described CT absorption could be a kind of signature for these encounter complex geometries and therefore the question arises whether the FLP literature has looked closely at all? However, at least one further FLP example should be added by the authors where such a CT absorption can be seen.

Page 3: Lines 94-95: "We have used these results to show for the first time that a higher concentration of active encounter complex in solution leads to faster small-molecule activation."

The fact that a higher concentration of the encounter complex leads to higher activity is well known in the literature (first described in Chem. Commun. 2008, 2130-2131, but also specifically here, for example:

<https://chemistry-europe.onlinelibrary.wiley.com/doi/full/10.1002/chem.202202273>: "Notably, an excess of the Lewis base has been found very favorable, which can be attributed to a higher concentration of the productive encounter complex." and here <https://pubs.acs.org/doi/full/10.1021>:

"To explain the above order of the piperidine...B(C₆F₅)₃ FLPs, one could, among other possibilities, assume on the basis of not too different FLP association energies decreasing FLP associations with increasing temperatures and consequently lowered actual FLP concentrations at higher reaction temperatures, which are anticipated to lead to reduced overall reaction rates.")

The authors should make this clear and emphasize the novelty value of their contribution and cite the relevant literature.

In summary, I support the publication of this paper in Nature Communications, provided that the authors can convincingly resolve the above mentioned issues, especially adding at least one further FLP example, where such a CT absorption can

be seen.

Version 1:

Reviewer comments:

Reviewer #1

(Remarks to the Author)

The authors addressed all comments imho satisfactory, by even doing extensive additional experimental work. Their rebuttal is strong, and the revised mss as well.

My only comments relate to the SI:

-A total of 4 different DFT methods are used in addition to the semi-empirical XTB method. I think it would be good to explain in the SI why the different methods were chosen for each step.

In addition, two small points about the calculations:

-on page S31 it is hidden that all calculations were performed with Gaussian 16, but does this also apply to the XTB? And the references to the program Gaussian and for the functionals and basis sets are missing.

-Figure S4 compares the CT bands of the different phosphines, but due to the different B(C₆F₅)₃ concentrations, a qualitative comparison of which phosphine gives the most intense CT band is not easily possible. It would therefore be nice if an additional figure is added with UV-vis spectra of the 4 new phosphines and PMe₃ with equal concentrations, e.g. UV-vis spectra of 50 mM equimolar PR₃-B(C₆F₅)₃ solutions.

Reviewer #2

(Remarks to the Author)

The authors have nicely and convincingly addressed all comments, which has further improved the quality of their manuscript. I have no further comments and therefore recommend publication in Nature Communications.

Reviewer #3

(Remarks to the Author)

All my questions were answered convincingly by the authors and my comments and suggestions were carefully considered. The revised manuscript has gained significantly in quality and in my opinion can be published in Nature Communications in its current form.

Version 2:

Reviewer comments:

Reviewer #1

(Remarks to the Author)

The authors have adequately addressed the final points raised by reviewer 1

Response to Reviewers' Comments

Manuscript ID: NCOMMS-23-55923

Original title: The Active Encounter Complex in Frustrated Lewis Pair Chemistry

Revised title: Quantifying Interactions in the Active Encounter Complex of Frustrated Lewis Pairs

We thank the reviewers for their time and detailed feedback for our manuscript. We have systematically numbered and addressed each comment raised below, and highlighted all changes in the manuscript. This has taken significantly more time than we first envisioned, but we believe the resulting manuscript is now much stronger thanks to the reviewers' feedback.

Reviewer #1 (Remarks to the Author):

The manuscript of Jupp, Barendt and coworkers presents the experimental determination of K_a for the benchmark frustrated Lewis pair $\text{PMes}_3 / \text{BCF}$. As such, the work is an essential contribution to the field of main group chemistry and catalysis. Yet, as the novel findings concerns merely the establishment of K_a , the manuscript is imho not suited for the Nature portfolio.

RESPONSE: We thank the reviewer for highlighting the importance of K_a determination for the FLP system to main-group chemistry and catalysis. We believe the additional work below demonstrates that this is not merely about establishing the K_a , but why the findings of this paper are important for the activation of H_2 .

1. line 9: the authors claim precious metal systems are environmentally unfriendly. Can the authors support this claim with scientific findings or rephrase this sentence?

RESPONSE: There is a lot of evidence that details the significant environmental cost of mining precious metals. However, we acknowledge that this is a complicated area, as the low catalyst loadings and (often) aqueous media used for precious metal catalysts can outweigh the benefit of using earth-abundant metals or other elements. As this sentence appears in the abstract and we don't have space for a nuanced discussion we have chosen to remove this part of the sentence in line with the reviewer's feedback, so the opening line reads: "Sustainable catalysts based on main-group elements, such as frustrated Lewis pairs (FLPs), have emerged as alternatives to precious metal systems."

2. l12: "acid, base" I would recommend using the Lewis acid, Lewis base nomenclature to be more precise.

RESPONSE: We have added these terms to the text in the manuscript.

3. l14: "However, there is no experimental methodology to study the active encounter complex". The authors highlight literature examples that showcase NMR spectroscopy (l58, l79) and UV-vis spectroscopy (l89) has been used as an experimental tool to characterise and study the encounter complex in FLP chemistry. I suggest the authors rephrase this claim.

RESPONSE: We have rephrased the sentence to provide further clarity that we are probing and quantifying the correct geometry of the FLP for small-molecule activation. The sentence now reads: "However, there is no experimental methodology to study and quantify the pre-associated complex that is specifically in the correct orientation for small-molecule activation, i.e. the *active encounter complex*."

4. I91: "This strategy provides the first experimental probe for the "active encounter complex" of an FLP in solution". I am not in favour of this claim given the cited literature examples on NMR and UV-vis spectroscopy. Can the authors rephrase this? The authors are indeed the first to quantify K_a , yet qualitative evidence for the encounter complex has been provided before.

RESPONSE: Although we believed we were precise in defining the "active encounter complex" as the orientation in which the acid and base are pointing towards each other, and describing the previous evidence provided by NMR and UV-vis spectroscopy for non-directional interactions in FLPs, we appreciate there could still be some confusion. We have therefore rephrased the statement to read: "This strategy provides the first experimental probe to quantify the association in the "active encounter complex" of an FLP in solution..."

5. I130: "very slightly exothermic process" I suggest to replace exothermic with exergonic, which is more appropriate term for a change in Gibbs free energy.

RESPONSE: Thank you for this correction, we have edited this text in the manuscript.

6. I201: "borane, but these three orientations are relatively higher in energy than the other orientations". Can the authors add the relative energies in the main text to provide the reader directly insight (e.g. Boltzmann) if the higher energy states are populated or not?

RESPONSE: The relative binding energies for these three structures have now been added to the manuscript, along with the value for the "face towards" complex for added clarity. The molecular IDs for these three orientations have been added in the SI to allow the reader to easily look up additional data in the computational spreadsheet if they so wish.

7. general question: is the conformation associated with the CT band the only "active encounter complex in solution" or can other conformations (many are calculated) also directly engage in a productive reaction with substrates like dihydrogen? if yes, then the quantification of K_a is directly coupled to the concentration of the CT structure, yet a higher concentration of "active encounter complex in solution" is present.

RESPONSE: This is an interesting question. The phosphine lone pair needs to be pointing towards the formally vacant p orbital on boron for productive reactions to occur. There will evidently some degree of flexibility to this, it won't be static in solution as it is in a single DFT calculation, and this is also reflected by the broad nature of the experimentally determined CT-band, but these are minor variations on the same general orientations. There will also be a degree of movement once H_2 is in the cavity of the encounter complex as well, as really the sigma bonding orbital on H_2 needs to overlap with the p orbital on the

borane. Additional text has been added to the SI in the “Time-dependent DFT (TD-DFT) calculations” section to explain this further.

8. for hydrogen activation experiments “active encounter complex” concentrations are used of 1.2% (1:1; A) and 11.1% (10:1; B), which is a ~10 fold increase. The authors use two approaches to quantify the reaction progress. 1. The UV-vis determination of the unreacted (CT) encounter complex concentration, 2. The determination of the H₂-activated product by ³¹P, ¹⁹F, and ¹¹B NMR analysis. After 60 min, quantitative ³¹P NMR indicated 23% (A) vs 41% (B) was formed. So my question is: given the tenfold increase in active FLP concentration and, likely, second order kinetics ([active encounter complex][H₂]), can the difference in product yield be understood? Intuitively, I would expect a bigger difference in yield.

RESPONSE: This result was a slight surprise to us as well. Our current hypothesis is that once some of the product has formed (the salt [HPMes₃]⁺ [HB(C₆F₅)₃]⁻), the cation or anion can interact with the remaining free phosphine or borane which will further disrupt the active encounter complex and hamper subsequent reactivity. This would also explain why the colour is lost in the UV-vis experiments at a faster rate than H₂ activation as measured by NMR spectroscopy. This will need a lot more experiments to fully probe, and is outside the scope of this current manuscript.

Reviewer #2 (Remarks to the Author):

Littlewood and co-authors have developed an experimental procedure that allows to determine the association constant for the active encounter complex of so-called frustrated Lewis pairs, with relatively high accuracy. They validated their results by a computational screening of all possible FLP complex configurations and funneling down to the only plausible candidate based on spectroscopic data, corresponding to the active complex.

This work is of high relevance to the field, since it gives experimental evidence of the existence of such pre-association of acid and base, as was already predicted based on computational grounds before. The authors also confirmed that the presence of these complexes is directly related to H₂ activation rates, thus indicating that tuning of the strength of association will impact the catalytic behavior.

RESPONSE: We thank the reviewer for their kind words about the impact of this work.

1. The proposed methodology looks very sound, both experimentally and computationally. As it was shown before in literature that different orientations of acid and base can co-exist, the authors followed the right track by using grid search and scanning methods to identify all plausible configurations, first by semi-empirical method and later fine-tuned by DFT. However, what I was missing was the highly relevant entropic effect. When accounting for the entropic contribution, does the computational binding free energy still agree with the experimentally determined exergonic association process?

RESPONSE: This is an interesting question that took a fair degree of computational resources to answer! We have carried out this investigation, and added the following text to the manuscript along with accompanying data in the SI:

“To probe entropic contributions, a cut-off at -10 kcal mol⁻¹ in Fig. 4c was applied, and 13 P(mes)₃/B(C₆F₅)₃ binding configurations were therefore selected and further investigated using the M06/6-311G(2df,p) level of theory with the D3 version of Grimme’s dispersion

correction.⁴⁸ As shown in Fig. S45, the entropic effect decreases the stability of the associated phosphine/borane by approximately 20 kcal mol⁻¹ (ranging from -18 to -23 kcal mol⁻¹), resulting in ΔG values ranging from -3 to 2.5 kcal mol⁻¹ (at 301 K). Six structures exhibit $\Delta G < 0$, with the 'face toward' configuration (Fig. 2a) having the lowest ΔG , though it is only 1.8 kcal mol⁻¹ more stable than the other five configurations. This suggests that the 'face toward' configuration co-exists with other configurations in solution, with no single dominant orientation, although it is significant that the active encounter complex in the P(mes)₃/B(C₆F₅)₃ FLP is the most stable orientation."

2. In literature, for this particular FLP, not all studies agreed on its favorability. Could the authors give an estimation of the Gibbs free binding energy, or at least comment on it?

RESPONSE: The experimental value of ΔG for this system is given on line 116-117 for the titration experiment (derived from the K_a). Computational ΔG values have now been calculated and added in lines 225-234 in the computational section; further details can be found in the SI.

3. Somewhat related to that, can the authors clearly mention in the main manuscript at what temperature the experiments were conducted?

RESPONSE: This omission has been corrected and added in both the main manuscript and SI with relevant ΔG calculations.

4. Finally, it is a pity that only a single FLP was considered for the study. If not in this work, it would be nice to see whether trends in computationally calculated strengths are conserved with this experimental approach.

RESPONSE: We have taken this comment to be a rather pivotal one to investigate further, and this has taken a considerable amount of time. A scope of arylphosphines was chosen to identify further CT-bands in related FLP systems. A total of four additional phosphines (P(*o*-tol)₃, P(xyl)₃, P(dur)₃ and P(C₆Me₅)₃) were synthesised and purified, which required another author (Dr Laura English) to be added to the project and manuscript. The four phosphines did give charge-transfer bands in combination with B(C₆F₅)₃, but the CT-bands were weaker than the P(mes)₃/B(C₆F₅)₃ combination. Very promisingly, the CT-bands did also increase in intensity with increasing phosphine:borane ratio, consistent with our hypothesis and the generality of this approach. Unfortunately, the weak CT-bands necessitated a higher concentration of the phosphine/borane mixture, which meant we encountered solubility limits of the phosphine at higher phosphine:borane ratios. This precluded us from obtaining a binding isotherm large enough to be able to accurately fit and calculate a K_a value from for these systems. However this is a technical limitation and not a fundamental flaw with the proposed methodology – a more sensitive UV-vis spectrophotometer or a cuvette with a longer pathlength would circumvent this issue, however these solutions are not currently available on the timescale and budget of this initial publication. This will form the basis of future work in the group, as this method enables us to compare fundamental properties like the change in encounter complex concentration (and FLP reactivity) as a function of solvent. A large paragraph of discussion and a new figure (Fig. 6) have been added to the manuscript discussing the above findings, and all the necessary experimental details and data have been

added to the supporting information. We believe this has really strengthened the manuscript and concept.

Reviewer #3 (Remarks to the Author):

The authors present a very accurate determination of the equilibrium constant K_a using sophisticated, state-of-the-art UV/VIS titration techniques including a triple determination for the association of probably the most studied FLP system in the literature, $B(C_6F_5)_3$ and $PMes_3$. Quantum chemical calculations were used to complement the absorption measurements. The employed methodology is sound even if the computational protocol could still be improved (but this would most likely not change the qualitative conclusions). The work can be fully reproduced by the provided details and data in the manuscript/supporting information and the data interpretation is precise.

RESPONSE: We thank the reviewer for their kind words about the study and the details provided in the supporting information.

1. Especially the UV/VIS part of the work is excellent, but the applicability of this approach is somewhat limited to those FLPs that also show UV/VIS absorption. This raises the question of how many of the (active) FLPs actually do so. The authors should add an analysis in order to be able to evaluate what impact the presented approach could actually have in practice in the search for better FLPs etc. (if necessary, the statement would then have to be weakened somewhat).

RESPONSE: This comment was addressed in our response to comment #4 from Reviewer 2. We have also amended our title and abstract somewhat to clarify our findings.

2. Nevertheless, this work is very valuable, as there are practically no reliable methods to characterize these flexible systems in solution (even if the sentence in the abstract "However, there is no experimental methodology to study the active encounter complex, ..." should be somewhat weakened (the authors mention also other experiments in the paper, e.g. using DOSY).

RESPONSE: This comment has been addressed in our response to comment #3 from Reviewer 1.

3. Likewise here: Page 3, lines 91-92 "This strategy provides the first experimental probe for the "active encounter complex" of an FLP in solution,...". When I do a quick literature search, I come across other experimental approaches to characterize the active encounter complex, see e.g. <https://doi.org/10.1039/D2RE00123C>. Here the authors should use weaker formulations.

RESPONSE: We have clarified this statement, as this comment relates to comment #4 from Reviewer 1. In response to the specific paper cited by the reviewer, this work is very elegant but it is only possible for Lewis adducts, and not systems that are fully frustrated (i.e. an encounter complex). There are many ways of analysing the strength of Lewis adducts and

then relating that to the relative Lewis acidity/basicity (another example is here: <https://chemistry-europe.onlinelibrary.wiley.com/doi/10.1002/chem.202003916>).

There was a recent paper published since the submission of this article that used microwave dielectric spectroscopy to assess the interactions of Lewis acids and bases, though this cannot be used for determining key thermodynamic information such as association constants, so our manuscript is still unique in being able to determine that. The following text and the corresponding reference has been added to the manuscript to draw attention to this: "Furthermore, during the revision of this manuscript, an elegant study using microwave dielectric spectroscopy to assess the interaction of acids and bases in solution was published, though it is not possible to determine association constants from these data.³¹"

4. The described CT absorption could be a kind of signature for these encounter complex geometries and therefore the question arises whether the FLP literature has looked closely at all? However, at least one further FLP example should be added by the authors where such a CT absorption can be seen.

RESPONSE: This comment has been addressed in our response to comment #4 from Reviewer 2; we have added four new systems that all exhibit CT-bands that increase in intensity as a function of phosphine:borane ratio.

5. Page 3: Lines 94-95: "We have used these results to show for the first time that a higher concentration of active encounter complex in solution leads to faster small-molecule activation. The fact that a higher concentration of the encounter complex leads to higher activity is well known in the literature (first described in Chem. Commun. 2008, 2130-2131, but also specifically here, for example: <https://chemistry-europe.onlinelibrary.wiley.com/doi/full/10.1002/chem.202202273> : "Notably, an excess of the Lewis base has been found very favorable, which can be attributed to a higher concentration of the productive encounter complex. " and here <https://pubs.acs.org/doi/full/10.1021>: "To explain the above order of the piperidine...B(C6F5)₃ FLPs, one could, among other possibilities, assume on the basis of not too different FLP association energies decreasing FLP associations with increasing temperatures and consequently lowered actual FLP concentrations at higher reaction temperatures, which are anticipated to lead to reduced overall reaction rates.") The authors should make this clear and emphasize the novelty value of their contribution and cite the relevant literature.

RESPONSE: We thank the reviewer for pointing out these missing references. We have removed the "for the first time" from the sentence highlighted above to now read: "We have used these results to show that a higher concentration of active encounter complex in solution leads to faster small-molecule activation." We have also added a sentence to the results section clarifying these findings with both these references included.

Changes to the SI

Following changes made in view of reviewer comments, the following figures have been added into the SI:

- Fig. S4 Comparison of charge-transfer bands of various triarylphosphines with $B(C_6F_5)_3$ in toluene
- Fig. S9: Stacked overlay of titration of $P(dur)_3/B(C_6F_5)_3$ in toluene.
- Fig. S10 1H NMR spectrum of $P(mes)_3$
- Fig. S15 1H NMR spectrum of $P(o\text{-tolyl})_3$
- Fig. S16 $^{31}P\{^1H\}$ NMR spectrum of $P(o\text{-tolyl})_3$
- Fig. S17 1H NMR spectrum of $P(xyl)_3$
- Fig. S18 $^{31}P\{^1H\}$ NMR spectrum of $P(xyl)_3$
- Fig. S19 1H NMR spectrum of $P(dur)_3$
- Fig. S20 $^{31}P\{^1H\}$ NMR spectrum of $P(dur)_3$
- Fig. S21 1H NMR spectrum of $P(C_6Me_5)_3$
- Fig. S22 $^{31}P\{^1H\}$ NMR spectrum of $P(C_6Me_5)_3$
- Fig. S45 Tabulated and graphical comparison of ΔE and ΔG of the $P(mes)_3/B(C_6F_5)_3$ pair

All pre-existing SI figures have been renumbered accordingly, with some captions having minor changes for clarity. Additional text has been added for the synthetic procedures for the additional chemicals added to the study.

Manuscript ID: NCOMMS-23-55923A

Title: Quantifying Interactions in the Active Encounter Complex of Frustrated Lewis Pairs

We thank all three reviewers for their feedback for our manuscript. All changes have now been made to the supporting information as requested by reviewer 1, which have resulted primarily in additions to the supporting information (a new figure, references, and explanatory text). The manuscript has only been amended by updating the figure numbers for figures referred to in the supporting information.

Reviewer #1 (Remarks to the Author):

The authors addressed all comments imho satisfactory, by even doing extensive additional experimental work. Their rebuttal is strong, and the revised mss as well.

RESPONSE: Thank you for the kind words, and for the further suggestions to improve the manuscript.

My only comments relate to the SI:

-A total of 4 different DFT methods are used in addition to the semi-empirical XTb method. I think it would be good to explain in the SI why the different methods were chosen for each step.

RESPONSE: Thank you for the suggestion – comments have now been added to the SI explaining the choice of methods.

In addition, two small points about the calculations:

-on page S31 it is hidden that all calculations were performed with Gaussian 16, but does this also apply to the XTb?

RESPONSE: The XTb optimisations were performed using the XTb software. We thank the reviewer for recommending the clarification, which has been added to the SI with the appropriate reference.

And the references to the program Gaussian and for the functionals and basis sets are missing.

RESPONSE: References 15-25 have been added to the SI to address this.

-Figure S4 compares the CT bands of the different phosphines, but due to the different B(C₆F₅)₃ concentrations, a qualitative comparison of which phosphine gives the most intense CT band is not easily possible. It would therefore be nice if an additional figure is added with UV-vis spectra of the 4 new phosphines and PMe₃ with equal concentrations, e.g. UV-vis spectra of 50 mM equimolar PR₃-B(C₆F₅)₃ solutions.

RESPONSE: As suggested, we collected three additional samples for the P(*o*-tol)₃, P(xyl)₃, and

P(mes)₃ systems with B(C₆F₅)₃ at 20 mM to correlate to the P(dur)₃ and P(C₆Me₅)₃ systems that used 20 mM B(C₆F₅)₃. This figure has been input as Fig. S5 and now allows for simpler quantitative comparison between the different systems. All subsequent SI figures, and any SI references in the main manuscript, have been re-numbered accordingly.

Reviewer #2 (Remarks to the Author):

The authors have nicely and convincingly addressed all comments, which has further improved the quality of their manuscript. I have no further comments and therefore recommend publication in Nature Communications.

Reviewer #3 (Remarks to the Author):

All my questions were answered convincingly by the authors and my comments and suggestions were carefully considered. The revised manuscript has gained significantly in quality and in my opinion can be published in Nature Communications in its current form.

RESPONSE: We thank reviewers 2 and 3 for their continued assessment of our manuscript and their kind words.